# A Settings and Systems Approach to Promoting the Health and Wellbeing of People with an Intellectual Disability

**DOI:** 10.3390/ijerph21040409

**Published:** 2024-03-28

**Authors:** Andrew Joyce

**Affiliations:** Centre for Social Impact, Swinburne University of Technology, Hawthorn, VIC 3122, Australia; ajoyce@swin.edu.au

**Keywords:** intellectual disability, health promotion, settings, workplace

## Abstract

Research has shown that people with an intellectual disability have higher rates of certain preventable health conditions and a higher rate of avoidable mortality relative to the general population. With respect to health behaviours, they also have lower levels of physical activity and poorer nutrition. Despite the increased health needs, this population cohort is less likely to have the opportunity to participate in health promotion programs. The interventions that have been delivered are programmatic and individual in focus and have lacked a broader ecological and settings focus, which makes them very difficult to scale. Health promotion programs designed for the general population, such as lifestyle campaigns, rarely reach people with an intellectual disability. This increases the importance of ensuring that the settings in which they live and engage with are as health promoting as possible. Interventions have been particularly absent in the workplace for people with an intellectual disability. This paper highlights gaps in a settings-and-systems-based approach to promoting the health and wellbeing of people with an intellectual disability, particularly with respect to workplace health promotion. The paper concludes with recommendations for a systems approach that integrates approaches across multiple settings to better promote the health of this population cohort.

## 1. Introduction

Due to different measurement criteria that have changed over time and different interpretations of classifications, precise estimates of the number of people with an intellectual disability are difficult to determine accurately [1]. Traditionally, intellectual disability was defined solely with regard to low intellectual functioning based on the standardised measurement of an IQ test; but more recently, the categorisation has broadened to impairment in intellectual function alongside limitations in daily activities [1]. Australian figures from 2003 indicate that estimates can vary between 1.6% of the population and 2.7% of the population, depending on the activity limitation classification used [1]. This variation depends on the sampling characteristics with respect to age (whether children under school age are included, for example), what cut-off criteria are used for the IQ score, and what other criteria are used [1]. Prevalence is also higher in males compared to females [1]. Worldwide, the estimated prevalence of intellectual disability is 1% [2]. Research has shown that people with an intellectual disability have higher prevalence rates of certain preventable health conditions including diabetes [3], obesity [3,4], and a higher rate of avoidable mortality relative to the general population [5]. There have been mixed findings in terms of cardiovascular risk factors, with research finding that women with an intellectual disability have higher rates of hypertension [6], but with more recent research indicating a lower risk profile for cardiovascular disease relative to population norms [3]. People with an intellectual disability are also more likely to have a lower rating of overall health and an increased rate of mental health problems [7].

With respect to health behaviours, the research indicates lower levels of physical activity and poorer nutrition; however, more positively, they have reduced rates of smoking and alcohol consumption relative to population norms [6,8,9]. Despite the increased health needs, this population cohort is less likely to have the opportunity to participate in prevention and health promotion programs [10]. There has also been relatively little health promotion research and evaluation with this population cohort [11]. A systematic review of physical activity interventions for people with an intellectual disability only found five relevant articles [8]. The focus of the interventions that were reviewed was largely programmatic, although there was some commentary on broader policy and settings approaches. A systematic review of nutrition interventions for people with an intellectual disability did uncover more articles, with 44 papers meeting the inclusion criteria [12]. However, the majority of interventions were focused on the individual level, and only four out of forty-four intervention studies had a settings component of change to the physical environment, three in a school setting and only one in a community residential setting.

Health promotion interventions for people with intellectual disabilities have tended to be programmatic and individual in focus and have lacked a broader ecological and settings focus [13]. There have been recommendations for more settings-based research and approaches for people with an intellectual disability [8,13,14,15]. Health promotion programs designed for the general population, such as lifestyle campaigns, often do not reach people with an intellectual disability, and there are few specific campaigns for this population cohort [12,15]. This increases the importance of ensuring that the settings in which they live and engage with are as health promoting as possible [15]. The purpose of this paper is to review the best practice approaches to settings-based health promotion, particularly in workplace settings, and to explore how this approach can be of benefit to people with an intellectual disability.

## 2. Settings Approaches

Health promotion has been implemented in settings for many years. Settings often have some sort of structure, culture, policies, and institutional values that can influence health behaviour [16]. Some of the common settings for health promotion include the workplace, schools, neighbourhoods or communities, and primary health care and hospitals [17,18]. It is important that a health promotion settings approach is inclusive of people with an intellectual disability in these community settings [14,15]. Whitelaw et al. [18] discussed the different elements of a settings-based approach and showed that there were different ways in which a settings-based model has been used. The most conservative approach uses the settings as a means to access populations for the delivery of individual-based programs. The ecological or ‘comprehensive/structural’ approach addresses the culture and structure of the organisation to promote health. In between these two positions are various combinations of individual and ecological approaches.

When considering the ecological perspective, organisational settings provide a way to focus on the determinants of health beyond personal control. They provide a good middle ground between individual behaviour and higher levels of social organisation. When working within settings it is important to recognise that settings often have multiple roles and functions. For example, settings such as schools and hospitals are not only organisations that provide for students and patients, respectively, but they are also workplaces and, for some people, homes (for example, boarding schools and residents of longer-term care facilities and nursing homes). Therefore each ‘type’ of setting may also perform the function of other settings. Coordination of efforts across settings and the integration of interventions in multiple settings are widely advocated. This is because people move in and out of settings in the course of their daily lives. In addition, it is recognised that interventions can work within several settings to maximise effectiveness [19].

With respect to settings-based work with people with an intellectual disability, there has been some work in supported accommodation. A scoping review of health promotion interventions in supported accommodation found that health education and exercise programs were the most common intervention types [20]. While the results were mixed, it was concluded that there were some promising findings with interventions that delivered health education for supported accommodation staff. It was also concluded that there needs to be more codesign with people with disabilities, which seemed to be lacking in the studies reviewed. Of further interest is that while there were some interventions that focused on integrating intervention components within the normal routines and procedures of the accommodation settings [20], for the most part, the interventions were programmatic in focus and were delivered in settings rather than taking an explicit settings focus. Thus, they were not structural and ecological to the full extent that is recommended for a comprehensive settings-based approach.

There has been some helpful research on the principles required for a settings-based approach for people with an intellectual disability [15]. These principles included ensuring the home and community environment is accessible and enabling for health. Key among these principles is the important role of care providers in creating an empowering environment and ensuring these care providers have the capacity themselves to create a health promoting environment [13]. Part of this involves managing the tension between enabling autonomy of choice over diet and also ensuring that healthy choices can be made [13]. For people with an intellectual disability, their support network involves care workers, and they form a key part of the “Healthy Settings for People with Intellectual Disabilities (HeSPID)” framework developed, which centres on ‘People, Places, and Preconditions’ [13]. This research is largely based on home and community settings, and while the principles and framework could be applied to workplaces, this was a not a focus of the research. Many people with an intellectual disability are employed in various types of workplaces, and this is another important health promotion setting.

## 3. Workplace as a Health Promotion Setting

Work conditions have a significant potential to influence health in either a positive or negative way, due to the amount of time spent at work [21]. Many of the early workplace health promotion programs, from the 1970s onwards, focused on promoting fitness through the provision of corporate fitness programs and providing facilities. This was followed by a focus on individual health issues such as weight control, cardiovascular disease risk appraisals, stress management, and ‘quit’ smoking programs. The focus of these programs was on individual behavioural change strategies, commonly as a component of screening, educational, or counselling programs [22]. The workplace provides an infrastructure and organization for coordinating and developing programs. This environment allows for health promotion messages to be efficiently and effectively communicated at a minimal cost [23,24,25].

The main limitation of this style of workplace health promotion programs is the traditional focus on behavioural programs, particularly those targeting individuals. This type of program supports the theory that the workplace is a convenient place to implement health promotion programs as opposed to being a setting that is involved in developing a program. Behavioural workplace health promotion approaches do not effectively deal with the social and economic determinants of health that are emphasised in the Ottawa Charter and other relevant workplace health promotion guides [26,27]. Creating healthy environments in workplaces and other settings together with strong community action are key aspects of the Ottawa Charter, which are missed if only taking a behavioural approach [26]. Work itself should be a ‘source of health’ [26]. In addition, even after controlling for lifestyle differences, there remains a significant gradient in health outcomes across occupational hierarchy. Behavioural changes tend to be short term unless there are concurrent changes to the social and cultural context that shapes an individual [25].

The comprehensive approach to workplace health promotion suggests that instead of the workplace being used simply as a good location for health promotion practitioners to implement programs, workplace environmental change needs to occur, which is instigated in partnership between staff and managers. This involves adopting multiple strategies that aim to improve the health status of employees and the population as a whole [23]. Over the last few years there have been a number of reviews of workplace health promotion covering mental health, nutrition, and physical activity that all conclude that multi-component programs are more effective [28,29,30,31,32,33]. This includes incorporating a range of topics and a range of strategies inclusive of individually focused strategies and organisational change strategies [28,30,32,34]. Education-only interventions have shown to be ineffective [28,33]. There is good evidence that multi-component work health promotion interventions can improve a range of health and wellbeing indicators and that they can reduce absenteeism [29,30,32]. Another consistent recommendation from the different reviews is that interventions need to be adapted to suit different workplace contexts, which can make replication and scaling more challenging [30,34]. Finally, it was noted in one review that there was a lack of research conducted with different population groups [34]. One such population group that has been missed in workplace health promotion comprises people with an intellectual disability.

## 4. Employment and Wellbeing among People with an Intellectual Disability

Employment forms an important role in the lives of many people with an intellectual disability; however, there are scant health promotion interventions and research in this setting for this population group. The history of employment for people with intellectual disability in Australia has been characterised by various policy approaches. In the 1950s and 1960s, the predominant approach was segregation of people with disabilities and the funding of what was termed sheltered workshops. In subsequent decades, there was more emphasis on inclusion, and the Disability Services Act of 1986 established two broad types of employment services, open and supported employment services [35]. What the Act produced was a bifurcated model in which open employment was only an option if someone did not need any support [35].

In the 1990s, there started to be a preference for open employment, and supported employment services were labelled as ‘disability business services’ or ‘business enterprises’. This name changed to Australian Disability Enterprises (ADEs) in 2008. Further reform has taken place more recently, as Australian Disability Enterprise Services was discontinued as a government funded program in 2021 [36]. Data from 2022 revealed there were 477 ADEs in Australia that were being operated by 147 organisations, employing 16,000 people [37]. Many ADE organisations have attempted to reposition themselves as social enterprises in the employment landscape. Despite all this reform for inclusion, ADEs are considered a setting at risk for exploitation, violence, and abuse [38]. Current policy and service delivery has directed school leavers with an intellectual disability into ADEs as the first option, and as data have revealed, transition out of an ADE is very unlikely [38].

The Disability Royal Commission has recommended that open and inclusive employment settings should be the first option for school leavers, and there were varying opinions among commissioners as to whether ADEs should be phased out or significantly reformed [38]. While many ADEs are now self-referring as social enterprises, the Commission felt that they had not undergone sufficient reform to provide an inclusive and community facing workplace that had a diverse workforce and provided training and other opportunities to transition to open employment. They described workplaces with these attributes as ‘social firms’ [38]. Data from the NDIS revealed very little movement from ADEs to other employment opportunities. Data from 2020 revealed that only 4% of 15–24 year olds had changed from an ADE to open employment, while 3% moved from open employment to ADEs [39]. The data for those older than 25 showed even lower levels of movement to open employment. Only 1% of people aged over 25 years moved from ADEs to open employment, while 3% moved from open employment to ADEs [39]. These results mirror studies from other countries, which show very low employment transition rates for people with an intellectual disability [40].

An inclusive health promotion workplace approach could be one of the key areas for action to address the current employment barriers and challenges that people with an intellectual disability experience. Despite all the reform, there remain significant barriers to inclusive employment and, at least from a research perspective, a seeming lack of focus on wellbeing within the workplace. While there is very little health and wellbeing intervention research conducted in workplaces for people with an intellectual disability, there is an emerging area of research in understanding job satisfaction for this population cohort. There have been a few papers that have explored the application of job satisfaction models and evaluation tools for people with an intellectual disability [41,42,43]. Having the psychological needs of a sense of autonomy, connection, and a sense of competence met in the workplace is associated with higher levels of job satisfaction [41].

Some of this research has been limited by small sample sizes from single organisations and other sampling limitations preventing a thorough test of job satisfaction models for people with an intellectual disability [44]. A recent study conducted a larger study on job satisfaction using Job Demands–Resource theory [44]. The study took place with 554 workers from Spain from 19 different workplaces. Eleven of these were sheltered workshops, and eight were supported employment opportunities (more community focused). Job Demands–Resource theory is based on the similarly named Job Demands–Resources model and the interaction of the personal (physical and psychological), organisational, and social demands of the job together with the resources available at these various levels as well [45]. Research has shown how the interaction of these two elements (demands and resources) influences wellbeing and job satisfaction in the workplace [44]. Previous research has utilised this model to reveal that low job demands and high levels of social support from co-workers and supervisors are related to an increased quality of working life [42].

Flores et al. [44] conducted a range of job satisfaction and job demand and resource survey instruments including the well-known Job Content Questionnaire (JCQ) [46]. Survey items were modified in some instances, and the data were collected through interviews with modified response options. The results showed that overall job satisfaction was high among all participants; but interestingly, those in inclusive employment had higher levels of job satisfaction, work engagement, lower overall scores on job demands, and increased scores on job resources. Other research has not found differences in job satisfaction between sheltered and inclusive employment [47].

The results of Flores et al. [44] were similar to past research with analysis revealing that high psychological demands were related to increased exhaustion and lower job satisfaction. Previous research has also found that the relationship between job demands and job satisfaction is mediated by personality characteristics such as conscientiousness [47]. It was concluded in this study that considering personality factors is important when matching for tasks [47]. Conversely, higher levels of job resources are related to increased job satisfaction, which has also been found in previous research [47]. In the Flores et al. [44] study, support from supervisors was the single biggest predictor of job satisfaction, and co-worker support was also found to be important. Qualitative research has also revealed the importance of supervisor and co-worker support for a sense of connection and wellbeing in the workplace for people with an intellectual disability [48]. Flores et al. [44] concluded that enhancing social connections could be a focus of workplace interventions, but further qualitative research is required to understand in depth other factors that may be important in determining job satisfaction for people with an intellectual disability.

There has also been some work understanding the role of managers in the support of workplace wellbeing for employees with an intellectual disability [49]. Using in-depth interviews with managers, the goal of this research was to understand how workplace health promotion is delivered at various stages of needs assessment, planning, intervention delivery, and evaluation. While conceptually, the analytic focus of the research was at a programmatic level, it was interesting that a number of systems concepts emerged. The managers discussed the importance of a culture of continuous improvement with respect to evaluation and always checking in with employees on their perceptions of their roles and various interventions. It was also apparent that there was great flexibility in intervention delivery between the different case study workplaces, which highlights the needs to tailor intervention strategies to the unique employee and business context of the organisation. Finally, the importance of an empowerment and partnership approach between managers and employees was emphasised. Thus, while the research was focused on understanding intervention delivery and evaluation, it did reveal some aspects of the culture of the organisation that were important for workplace health promotion for this cohort. Further research is required to more explicitly detail the structural and cultural elements of the workplace that foster wellbeing. It is the structure and culture of a workplace itself rather than just the intervention delivery that is important to understand [50].

The other interesting area to emerge from this research was the topic focus of the interventions. While this was not made explicit, it would seem that job satisfaction and social connection were the dominant focuses given some of the intervention strategies [49]. There was no mention of physical activity or nutrition, even though they are areas of need and have been targets of workplace health promotion interventions for the general population. Thus, there are many gaps with respect to workplace health promotion for people with an intellectual disability. The emerging research indicates that the quality of the work experience determines the levels of job satisfaction and sense of connection. This mirrors research in the workplace setting with other vulnerable cohorts in that it is not just having employment that is important for wellbeing but the quality of the workplace experience [50]. What is lacking is intervention research that takes a comprehensive settings-based approach.

## 5. A Comprehensive Settings-and-Systems-Based Approach

There are a number of research gaps that currently exist with respect to a settings-based approach with people with an intellectual disability. While there is some evidence now on the characteristics of job satisfaction for people with an intellectual disability, there is little evidence, as far as we are aware, on intervention studies attempting to improve these factors. Further, there is no evidence, as far as we are aware, that has attempted to improve nutrition and physical activity within the workplace for people with an intellectual disability, despite this being an area of high need. Further, as reviewed earlier, many of the health and wellbeing interventions that have been delivered in other settings, particularly supported accommodation, are programmatic in focus and have not taken a settings-based approach. A comprehensive settings-based approach also needs to be coordinated across multiple settings such as the workplace, schools, supported accommodation, and community settings. Such an approach necessitates a systems orientation to health promotion delivery delivered with people with an intellectual disability.

Systems thinking is an approach that considers how different elements of a system (for instance different settings) connect with and influence each other [51]. A systems-based approach has many common elements with settings approaches, whereby the focus is on the changes to policies, routines, relationships, power structures, and values [52]. Addressing settings-based change has been recommended as a way to move away from the more limited approach of individual behavioural risk factors and to address higher levels of social organisation [18,19]. Key to this is understanding that settings are complex environments, and flexible approaches are required to address the culture and structure of a setting/system [53,54]. Whether there is positive change relies on how the intervention components interact with the system in which they are being delivered. That is, the same intervention component could have different results in different settings depending on how it is influenced by the people, culture, structure, and other elements within any one setting. Thus, is important to adjust interventions to suit the particular characteristics of the setting to ensure the best possible outcome [53,54].

Another important systems consideration is providing as many opportunities as possible for health behaviour change to occur. A health promotion practitioner needs to repeatedly provide a program with the aim of eventually producing a scenario where the information provided together with the psychological state of an individual produces a change effect, “hitting a lever point” [55]. This analogy could equally be applied at a socio-ecological level. The policy goal in this sense is to create as many healthy settings as possible to increase the likelihood of creating that requisite scenario for change. Addressing as many determinants and settings as possible, such as friends, family, neighbours, schools, workplaces, places of worship, community venues and groups, primary care, and media, increases the possibility of creating opportunities for healthy behaviour change.

The current research base for settings-based approaches for people with an intellectual disability resembles the criticisms made of settings-based approaches generally over twenty years ago. Settings level interventions tended to:

“*review single strategy interventions (health education in schools), single risk factor interventions (smoking cessation in workplaces) or single health impact measures, rather than exploring the total effect of a multi-focus health promotion approach across a range of outcomes*.”[56] (p. 17).

There has not been consideration of the broader relationships both within and between settings that are features of an ecological approach and considered single-issue outcomes [57]. In the last 20 years, there has been some improvement in the evidence base for health promotion settings aimed at the general population. Using a logic analysis approach, there is evidence that systems-based interventions can improve the health of workers, and this research can guide a routine approach to data collection [58]. A review of the workplace health literature commissioned by VicHealth concluded that the strongest evidence base was for workplace programs that addressed system level change such as communication and job redesign [59]. It was recommended that organisations take a systems-based approach to reducing stress in the workplace. Systems-based interventions targeting both nutrition and physical activity together, such as changes to canteen and food price, have also been effective in changing health behaviours [60]. Thus, workplaces can routinely track indicators for job stress, smoking, physical activity, and nutrition, where evidence based reviews have shown that these domains can be improved with well-planned interventions [59,60,61].

There have been a number of recommendations for systems approaches to health promotion, although there are not many practice examples to date [62,63]. However, it has been demonstrated that a coordinated approach to multiple settings (education, work, and community) can have a synergistic effect for the general population [63,64]. By having a common language and branding of settings work, practitioners can leverage existing partnerships to encourage personnel in other settings to engage in changes to promote wellbeing. Key to this approach is having a suite of resources that allows for flexibility in approach, whereby each setting can make the changes necessary to suit their particular context. We are not aware of any attempts to address multiple settings for people with an intellectual disability. There is a need for some kind of tool that can guide health promotion practice in workplaces and other settings that engage people with an intellectual disability. This same recommendation has been made for home and community settings [13]. What would be important is that these tools and guidance materials be complementary across the different settings. It cannot be assumed that settings guides for the general population will be relevant for people with an intellectual disability. They need to be purposely designed for these settings, taking into account the particular needs and environments, such as the important role of support staff [13].

## 6. Conclusions

There is a strong need for further intervention research that takes an integrated approach across various settings such as day programs, workplaces, and community settings [14]. Improvements in health require coordination and consistency across all these settings, and people with an intellectual disability need to be partners in the planning, delivery, and evaluation of health promotion initiatives [14]. For this to occur requires a systems approach whereby there is coordination across multiple settings. That way there is a consistent approach and language being used that can help to leverage change across multiple settings and start to generate the kind of systems change that is required to promote the health and wellbeing of people with an intellectual disability. Further research is required on the most appropriate systems model and tools to support this kind of practice. This includes considerations of adapting approaches to suit different setting contexts and enabling strong participation processes with people with an intellectual disability. Such work has the potential to improve the health of people with an intellectual disability in their place of work and other settings, which is important in addressing the health disparities this population group experiences.

## Data Availability

No new data were created or analyzed in this study. Data sharing is not applicable to this article.

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
