# Peer review of "A Settings and Systems Approach to Promoting the Health and Wellbeing of People with an Intellectual Disability"

_ijerph, 2024, doi:10.3390/ijerph21040409_

Round 1

Reviewer 1 Report

Comments and Suggestions for Authors

 This article describes a settings and systems approach to promoting the health and well-being of people with an intellectual disability.  The author contends that while individuals with disabilities have higher rates of preventable health issues and mortality and less optimal health behaviors compared to the general population, this group is less likely to receive health education programs.  Moreover, a broader ecological and systems approach is needed to promote health among this cohort of individuals with disabilities.  The review is as follows:

1.       The Introduction should be expanded to include a definition of ‘intellectual disability’. 

2.       Lines 26-27 – In “Figures from 2003 indicate that estimates can vary between 1.6% of the population to 2.7% of the population…”, specify the locale.  Is this within Australia?  Are there subgroup statistics (e.g., age, gender, race, etc.)?  Information on historical, political, social, cultural and other factors within the society can also help provide context to the reader.

3.       Lines 99-10 – In “There has been some terrific research on the principles required for a settings-based approach for people with an intellectual disability”, the use of the word ‘terrific’ seems slanted.  Perhaps other words can be considered (e.g., promising, helpful, valuable, helpful).

4.       Under Settings Approaches, there is an insightful description of an ecological or ‘comprehensive/structural’ approach.

5.       Lines 135-137 – In “Behavioral workplace health promotion approaches don’t effectively deal with the social and economic determinants of health that are emphasized in the Ottawa Charter and other relevant publications”, it may be helpful to provide examples of the social and economic determinants of health that are emphasized in the Ottawa Charter.

Overall, this is a unique and pertinent paper on a very important topic.  Consider expanding the introduction section to make the paper more compelling and providing more information to help contextualize the paper.

Author Response

  1. The Introduction should be expanded to include a definition of ‘intellectual disability’.

Response: This has been added.

  1. Lines 26-27 – In “Figures from 2003 indicate that estimates can vary between 1.6% of the population to 2.7% of the population…”, specify the locale.  Is this within Australia?  Are there subgroup statistics (e.g., age, gender, race, etc.)?  Information on historical, political, social, cultural and other factors within the society can also help provide context to the reader.

Response: This has been added.

  1. Lines 99-10 – In “There has been some terrific research on the principles required for a settings-based approach for people with an intellectual disability”, the use of the word ‘terrific’ seems slanted.  Perhaps other words can be considered (e.g., promising, helpful, valuable, helpful).

Response: Thanks this has been changed.

  1. Under Settings Approaches, there is an insightful description of an ecological or ‘comprehensive/structural’ approach.

Response: Thanks

  1. Lines 135-137 – In “Behavioral workplace health promotion approaches don’t effectively deal with the social and economic determinants of health that are emphasized in the Ottawa Charter and other relevant publications”, it may be helpful to provide examples of the social and economic determinants of health that are emphasized in the Ottawa Charter.

Response: Thanks for this tip this information has been added.

Overall, this is a unique and pertinent paper on a very important topic.  Consider expanding the introduction section to make the paper more compelling and providing more information to help contextualize the paper.

Response: Thanks for your helpful suggestions.

Reviewer 2 Report

Comments and Suggestions for Authors

Thank you for this interesting paper on interventions for workplace health promotion for people with an intellectual disability. I have just two minor amendments for you to consider:

1.     Section 3: Workplace as a health promotion setting. While this section provides some background to health promotion in the general population, I'm not sure it really provides anything of value in relation to the ID population. I would suggest reducing it to a few salient points and putting these into the section on employment and wellbeing among people with ID, perhaps at the start of the paragraph beginning with line 197 as a lead into health promotion in the workplace for people with ID.

2.     There are some grammatical errors in the paragraph beginning with Line 197 which needs reviewing.

Author Response

Thank you for this interesting paper on interventions for workplace health promotion for people with an intellectual disability. I have just two minor amendments for you to consider:

  1. Section 3: Workplace as a health promotion setting. While this section provides some background to health promotion in the general population, I'm not sure it really provides anything of value in relation to the ID population. I would suggest reducing it to a few salient points and putting these into the section on employment and wellbeing among people with ID, perhaps at the start of the paragraph beginning with line 197 as a lead into health promotion in the workplace for people with ID.

Response: I have edited down this section to make it shorter but have kept is a separate section. The advantage of keeping this as a separate section of the paper is to illustrate the difference between a comprehensive settings based approach in the workplace versus using the setting as a vehicle to deliver health education in the workplace. This follows the previous section of the paper and becomes an important theme for the rest of the paper particularly in the conclusion. I am concerned that by merging with the next section of the paper this point will get lost. It sets the scene for what should be expected in workplaces that have employees with an intellectual disability.

  1. There are some grammatical errors in the paragraph beginning with Line 197 which needs reviewing.

Response: This sentence has been rewritten.

Reviewer 3 Report

Comments and Suggestions for Authors

Thank you for the opportunity to review this paper. I really enjoyed reading it and I think the authors offer an important perspective on how to broaden our frame of reference related to health and wellbeing to include a setting and systems approach. I highlighted many statements throughout, just as a means to agree with important points provided!

I am recommending minor revisions:

1- can you please include a definition of intellectual disability in the opening paragraph?

2- 1st pg. line 31 - I don't consider injuries a health condition. Can you support why you included it under 'certain preventable health conditions'?

3- 1st page lines 38-39-  you use lower twice in that sentence - do you mean "and also"?

4- 1st page line 41 - People don't receive health promotion programs (although they may receive prevention). Could you replace receive with "participate in" or something along those lines?

5- Please include a purpose of the paper. To review... ect.

6- pg. 3 line 137 - Can you provide a reference for the Ottawa Charter and also provide examples of "other relevant publications"?

7- pg. 5 - general comment. I would love a bit more detail about the included studies (e.g., Flores et al. - where was this study conducted? With how many people?)

Author Response

Thank you for the opportunity to review this paper. I really enjoyed reading it and I think the authors offer an important perspective on how to broaden our frame of reference related to health and wellbeing to include a setting and systems approach. I highlighted many statements throughout, just as a means to agree with important points provided!

I am recommending minor revisions:

1- can you please include a definition of intellectual disability in the opening paragraph?

Response: This has been added.

2- 1st pg. line 31 - I don't consider injuries a health condition. Can you support why you included it under 'certain preventable health conditions'?

Response: Thanks for this suggestion. Injuries are certainly preventable and an important public health area but the paper doesn’t really address injuries as a topic and as such, this point has been removed.

3- 1st page lines 38-39-  you use lower twice in that sentence - do you mean "and also"?

Response: This sentence has been edited to make the point clearer.

4- 1st page line 41 - People don't receive health promotion programs (although they may receive prevention). Could you replace receive with "participate in" or something along those lines?

Response: Thanks for pointing this out. This sentence has been changed and also the wording in the abstract has also been changed.

5- Please include a purpose of the paper. To review... ect.

Response: This has been added to the paper.

6- pg. 3 line 137 - Can you provide a reference for the Ottawa Charter and also provide examples of "other relevant publications"?

Response: This has been added.

7- pg. 5 - general comment. I would love a bit more detail about the included studies (e.g., Flores et al. - where was this study conducted? With how many people?)

Response: This information has been added.

Round 2

Reviewer 1 Report

Comments and Suggestions for Authors

The revised manuscript is clearer and more comprehensive.  The author has done well in responding to the suggested feedback.  One item remaining is that it is not clear who comprises the study sample.  Lines 213-214 mention that “Much of this research has been limited by small sample sizes but a recent study conducted a larger study on job satisfaction using Job Demands-Resource theory [44]’.  It would be helpful to provide for the reader the demographic characteristics (e.g., gender, age, race, ethnicity. socioeconomic status) of the study samples used in the literature reviews to paint a clearer picture of the participants in these studies. Doing so would help strengthen the manuscript.

Author Response

The revised manuscript is clearer and more comprehensive.  The author has done well in responding to the suggested feedback.  One item remaining is that it is not clear who comprises the study sample.  Lines 213-214 mention that “Much of this research has been limited by small sample sizes but a recent study conducted a larger study on job satisfaction using Job Demands-Resource theory [44]’.  It would be helpful to provide for the reader the demographic characteristics (e.g., gender, age, race, ethnicity. socioeconomic status) of the study samples used in the literature reviews to paint a clearer picture of the participants in these studies. Doing so would help strengthen the manuscript.

Response: Thanks for this helpful suggestion. More information has been added about the limitations of the previous research and also the sentences have been moved to make this point clearer. It was not so much demographic sampling issues but sample sizes and sampling procedures that could enable the type of quant analysis required for the model building that these researchers were seeking.